# Biased Noise Kerr-Cat Qubits for Fault-Tolerant Quantum Computing

Shruti Puri

Department of Applied Physics, Yale University
* shruti.puri@yale.edu

July 26, 2021

## Abstract

**In this chapter I will introduce the bosonic Kerr-cat qubit which is encoded in a parametrically driven nonlinear oscillator. We will see that this qubit couples anisotropically with its environment which causes a structure or bias in the noise channel of the qubit. The structure of noise is preserved even during non-trivial gate operations which can be exploited for designing more efficient, fault-tolerant quantum error correcting codes.**

## 1 Introduction

Fault-tolerant quantum error correction (QEC) allows reliable execution of a quantum algorithm of arbitrary length using unreliable circuit components as long as the error rate is below a threshold. Over the past two decades a lot of effort has been directed towards increasing the threshold and decreasing the overhead requirements for fault-tolerant quantum computation. Most of this work is based on assuming generic noise models without

any structure. However, many qubit architectures have in-built protection against certain errors which leads to a highly asymmetric or biased noise channel. More importantly, if this protection persists during non-trivial gate operations then it becomes possible to design more efficient error correction protocols. In this chapter I will introduce the bosonic Kerr-cat qubit which has these properties and show how it can be used to improve the performance of error correcting codes.

## 2    QEC with Biased-Noise Qubits

Before starting discussion about QEC we must describe the noise that affects the underlying qubit hardware. A practical model for quantum noise is the independent and identically distributed Pauli noise model, in which each qubit is subject to the channel represented as,

$$\mathcal{E}(\rho) = p_0 I \rho I + p_{\mathrm{x}} X \rho X + p_{\mathrm{y}} Y \rho Y + p_{\mathrm{z}} Z \rho Z \tag{1}$$

In the above expression $p_{\mathrm{x}}, p_{\mathrm{y}}$, and $p_{\mathrm{z}}$ represent the probabilities of a Pauli $X, Y$, and $Z$ error respectively. The probability of no error is $p_0$ and in the absence of any leakage $p_0 = 1 - (p_{\mathrm{x}} + p_{\mathrm{y}} + p_{\mathrm{z}})$. Typically, the work in the field of quantum error correction (QEC) is based on qubits which have no special noise structure. That is, the qubit couples to the environment isotropically and all the Pauli errors are equally likely, resulting in what is known as a *depolarizing noise channel*. However, in many qubit architectures, such as superconducting fluxonium qubits [1], quantum-dot spin qubits [2,3], nuclear spins in diamond [4] etc., the coupling to the environment is anisotropic so that one type of error dominates over all the others. Such qubits are said to have a *biased noise channel* and for the discussion here we will assume that the dominant error is $Z$-type. In this case it is possible to define a quantity called the bias, $\eta$ which is the ratio of the probability of $Z$ error and the probability of $X$ and $Y$ errors,

$$\eta = \frac{p_{\mathrm{z}}}{p_{\mathrm{x}} + p_{\mathrm{y}}} \tag{2}$$

For the depolarizing noise channel $p_{\mathrm{x}} = p_{\mathrm{y}} = p_{\mathrm{z}}$ and $\eta = 0.5$. In the other extreme case, when $p_{\mathrm{x}} = p_{\mathrm{y}} = 0$, $\eta \to \infty$.

In the last few years, variations of the surface code have been developed which are highly effective against biased-noise errors [5–8]. However, note that the noise channel of an idle qubit can be very different from when gates are being implemented on it. This is especially important when biased noise is involved. For example, if the noise is $Z$-biased then $Z$ errors remain dominant after implementation of a diagonal gate. This is because $Z$ errors commute with the diagonal interaction Hamiltonian realizing the gate at all times. However, in general $Z$ errors may no longer be dominant after implementation of a non-diagonal gate like the controlled-not or CX gate. In the CX, for example, a $Z$ error in the target qubit does not commute with the underlying interaction Hamiltonian. In fact, in the standard CX, $Z$ errors in the target qubit during the gate propagate as a linear combination of $Z$ and $Y$ errors. It is also possible to have control errors which will cause over-rotation or under-rotation of the target qubit of the CX gate, inevitably leading to $X$ error in the data qubits. Unfortunately, this means that even if the idle noise channel of the qubit is biased towards $Z$-errors, the noise channel will become unbiased when a CX gate is implemented. In fact, a no-go result, first described in [9], forbids the existence of a bias-preserving CX gate between standard, strictly two-level qubits. Unfortunately, the CX gates are crucial for measuring the stabilizers of the surface code variants tailored

for biased-noise. In the absence of a bias-preserving CX, the performance of these codes deteriorate [10] and therefore it seems that there is no advantage to having biased-noise qubits. Fortunately, it has now been shown that it is possible to engineer a bias-preserving CX gate with bosonic-cat qubits.

A bosonic qubit is a qubit encoded in a two-dimensional subspace of an infinite-dimensional bosonic mode. Typically these qubits are defined in harmonic oscillators. However, an exception is the so called *Kerr-cat qubit* which is realized in a driven Kerr nonlinear oscillator [11]. Unlike their harmonic counterparts, a large Kerr-nonlinearity is desirable for a Kerr-cat qubit and it enables fast gate operations. In the following sections we will discuss the Kerr-cat qubit, its noise channel and the CX gate, which is the key component for error correction with biased noise.

## 3 The Kerr-Cat Qubit

The Hamiltonian of a two-photon driven Kerr-nonlinear oscillator in a frame rotating at the oscillator frequency $\omega_{\mathrm{r}}$ is given by

$$H_0(\phi) = -K a^{\dagger 2} a^2 + P(a^{\dagger 2} e^{2i\phi} + a^2 e^{-2i\phi}) = -K \left( a^{\dagger 2} - \alpha^2 e^{-2i\phi} \right) \left( a^2 - \alpha^2 e^{2i\phi} \right) + \frac{P^2}{K}. \quad (3)$$

Here, $\alpha = \sqrt{P/K}$, $K$ is the strength of the nonlinearity while $P$ and $\phi$ are respectively the amplitude and phase of the drive. For two coherent states $|\pm\alpha e^{i\phi}\rangle$, $a^2|\pm\alpha e^{i\phi}\rangle = \alpha^2 e^{2i\phi}|\pm\alpha e^{i\phi}\rangle$. Hence, it is clear that the even- and odd-parity cat states $|\mathcal{C}^\pm_{\alpha e^{i\phi}}\rangle = \mathcal{N}_\pm(|\alpha e^{i\phi}\rangle \pm |-\alpha e^{i\phi}\rangle)$ are the degenerate eigenstates of this Hamiltonian [11, 12]. Figure 1(B) shows the eigenstates of the oscillator in the rotating frame (see also [13] for a detailed discussion of the eigenspectrum.) The degenerate cat-subspace $\mathcal{C}$ (green) is separated from the rest of the Hilbert space $\mathcal{C}_\perp$ (orange) by a large energy gap, which in the limit of large $\alpha$ is well approximated as $|\Delta\omega_{\mathrm{gap}}| \sim 4K\alpha^2$. For large $\alpha$, the energy gap between pairs of even and odd parity excited states $|\psi^\pm_{e,n}\rangle$ decreases exponentially with $\alpha^2$ for $n \lesssim \alpha^2/4$ [13, 14]. As a result, the eigenspace of the two-photon driven oscillator reduces to $\alpha^2/4$ pairs of quasi-degenerate states. This Hilbert space symmetry is important for the exponential suppression of $X, Y$ errors [14]. Observe that in the limit $P \to 0$, the even and odd parity cat states continuously approach the vacuum and single-photon Fock states respectively. It follows that, starting from an undriven oscillator in vacuum (or single-photon Fock state) it is possible to adiabatically prepare the state $|\mathcal{C}^+_{\alpha e^{i\phi}}\rangle$ (or $|\mathcal{C}^-_{\alpha e^{i\phi}}\rangle$) by increasing the amplitude of a resonant two-photon drive at a rate $\ll 1/|\Delta\omega_{\mathrm{gap}}|$ [11, 13, 14]. Recently, this adiabatic preparation of cats was implemented in a superconducting circuit platform [15]. The phase $\phi$ of the two-photon drive is a continuous parameter that specifies the orientation of the cat in phase space. We define the cat-qubit with the phase $\phi = 0$ (see Fig. 1(A)) and for the discussion of the following two sections we will fix this phase. As we will see in a few sections, this phase degree of freedom becomes crucial for the implementation of the CX gate.

### 3.1 Relationship with The Dissipative-Cat

The Kerr-cat is realized via engineering the Hamiltonian of a Kerr-nonlinear oscillator. It is also possible to realize a cat-qubit by engineering a two-photon dissipation channel of a parametrically driven oscillator [16, 17]. Remarkably, these two seemingly different mechanisms can be employed in unison to design a better cat qubit [11, 13, 14]. To see this, consider the master equation of the parametrically driven Kerr-nonlinear oscillator

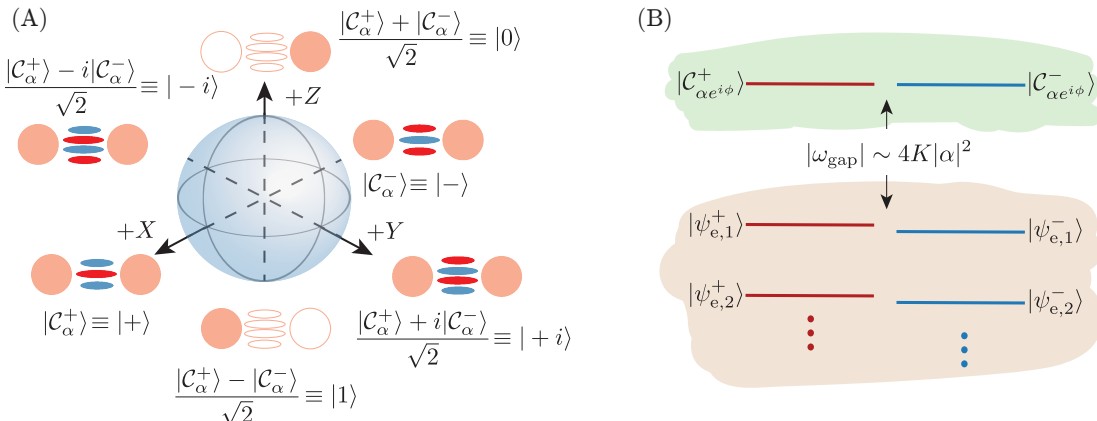

Figure 1: **(A)** Bloch sphere of the cat qubit. **(B)** Eigenspectrum of the two-photon driven nonlinear oscillator in the rotating frame (3). The cat states $|\mathcal{C}^{\pm}_{\alpha e^{i\phi}}\rangle$ are exactly degenerate and the rest of the Hilbert space can be divided into an even and an odd parity manifold. The cat subspace, highlighted in green, is separated from the first excited state by an energy gap $|\Delta\omega_{\mathrm{gap}}| \sim 4K\alpha^2$. The energy difference between the first $n \sim \alpha^2/4$ pairs of excited states (highlighted in orange) $|\psi^{\pm}_{\mathrm{e},n}\rangle$ decreases exponentially with $P$ or equivalently with $\alpha^2$. These excited state pairs are consequently referred to as quasi-degenerate states.

in presence of a white two-photon dissipation channel,

$$\dot{\rho} = -i[H_0(\phi), \rho] + \kappa_2 \mathcal{D}[a^2]\rho. \tag{4}$$

Here $\kappa_2$ is the rate of two-photon dissipation. Such a dissipation channel enforces that photons are lost in pairs to the environment. The dissipative dynamics can be understood in the quantum-jump approach in which the deterministic evolution governed by the non-Hermitian effective Hamiltonian $H = H_0 - i\kappa_2 a^{\dagger 2} a^2/2$ is interrupted by two-photon jump events. The non-Hermitian Hamiltonian is analogous to Eq. (3) with the Kerr-nonlinearity $K$ replaced by a complex quantity $K + i\kappa_2/2$. The cat states $|\mathcal{C}^{\pm}_{\beta}\rangle$ are degenerate eigenstates of the non-Hermitian Hamiltonian $H|\mathcal{C}^{\pm}_{\beta}\rangle = E|\mathcal{C}^{\pm}_{\beta}\rangle$ where $E$ is a complex quantity $E = P^2/(K + i\kappa_2/2)$ and $\beta = e^{i\phi}\sqrt{P/(K + i\kappa_2/2)}$. Moreover, the cat states are also eigenstates of the two-photon jump operator $a^2|\mathcal{C}^{\pm}_{\beta}\rangle = \beta^2|\mathcal{C}^{\pm}_{\beta}\rangle$. Therefore, the states $|\mathcal{C}^{\pm}_{\beta}\rangle$ are invariant to two-photon dissipation and are the steady states of the system. Recall that, we defined the cat qubit $|\mathcal{C}^{\pm}_{\alpha}\rangle$ using real and positive coherent state amplitude $\alpha$. Therefore, for this qubit to be the steady-state in the presence of two-photon dissipation, the phase and amplitude of the required two-photon drive must be $2\phi_0 = \tan^{-1}(\kappa_2/2K)$ and $P = \alpha^2\sqrt{K^2 + \kappa_2^2/4}$ respectively. There is a two-fold advantage to this hybrid Hamiltonian-dissipative design. Firstly, the large Kerr-nonlinearity (typically in the range of 10-100 MHz in superconducting circuits) allows fast, high fidelity gate operations ($\sim$100-1000 times faster than the decoherence time in superconducting circuits) [15]. Secondly, the two photon dissipation allows for autonomous correction of leakage by "cooling" the oscillator to the steady-state cat qubit manifold [14].

## 3.2 Coupling to External Degrees of Freedom: Coherent Dynamics and Noise Properties

It is possible to describe the coupling of the oscillator to an external degree of freedom using an operator of the form,

$$O = \chi_{m,n}(t)e^{i\omega_{\mathrm{r}}(m-n)t}a^{\dagger m}a^n + \mathrm{h.c}, \tag{5}$$

The expression above is written in a frame rotating at the resonance frequency $\omega_{\rm r}$ of the oscillator. We will also assume, without any loss of generality, that $m > n$. The total Hamiltonian of the system and external mode is $H_0(\phi) + O$. Next, we will further go in the interaction picture of $H_0(\phi)$,

$$\tilde{O} = \chi_{m,n}(t)e^{i\omega_{\rm r}(m-n)t}e^{iH_0(\phi)t}a^{\dagger m}a^n e^{-iH_0(\phi)t} + \chi_{m,n}^*(t)e^{-i\omega_{\rm r}(m-n)t}e^{iH_0(\phi)t}a^{\dagger n}a^m e^{-iH_0(\phi)t} \tag{6}$$

In order to understand the dynamics we can look at the matrix elements of $\tilde{O}$ in the eigenbasis of $H_0(\phi)$. We will denote the cat-subspace by $\mathcal{C}$ and the projector on the cat subspace as $P_{\mathcal{C}} = |\mathcal{C}_{\alpha e^{i\phi}}^+\rangle\langle\mathcal{C}_{\alpha e^{i\phi}}^+| + |\mathcal{C}_{\alpha e^{i\phi}}^-\rangle\langle\mathcal{C}_{\alpha e^{i\phi}}^-|$. Within this degenerate cat subspace and in the limit of large $\alpha$, $P_{\mathcal{C}}\tilde{O}P_{\mathcal{C}} \sim \alpha^{m+n}(\chi_{m,n}(t)e^{i\omega_{\rm r}t(m-n)}e^{i\phi(n-m)}+{\rm c.c.})P_{\mathcal{C}}$ for $m+n$ =even and $P_{\mathcal{C}}\tilde{O}P_{\mathcal{C}} \sim \alpha^{m+n}(\chi_{m,n}(t)e^{i\omega_{\rm r}t(m-n)}e^{i\phi(n-m)} + {\rm c.c.})Z_\phi$ for $m + n$=odd. Here $Z_\phi$ is the Pauli $Z$ matrix, $Z_\phi = |\mathcal{C}_{\alpha e^{i\phi}}^+\rangle\langle\mathcal{C}_{\alpha e^{i\phi}}^-| + |\mathcal{C}_{\alpha e^{i\phi}}^-\rangle\langle\mathcal{C}_{\alpha e^{i\phi}}^+|$. In writing this expression we have ignored terms which are exponentially small in $\alpha$ and hence this expression is a good approximation in the large $\alpha$ limit. We see that if the coupling has a non-zero spectral density at $\omega_{\rm r}(m-n)$ then the effect of $O$ is to primarily cause Rabi-oscillation around the $Z$-axis of the Bloch sphere (if $m+n$ is odd). To elaborate with a simple example, consider the situation when $\phi = 0$ and the oscillator is driven by an in-phase classical coherent drive of strength $\chi_0$ at frequency $\omega_{\rm r}$. In this case, $O = \chi_0\cos(\omega_{\rm r}t)(e^{i\omega_{\rm r}t}a^\dagger+{\rm h.c})$ and following the above argument we find that such a drive causes Rabi rotation at frequency $\Omega_{\rm z} = 2\alpha\chi_0$. In fact, such Rabi oscillations have been implemented in a recent experiment [15].

Observe that, if the oscillator started in one of the cat states then there is a non-zero probability of leakage to the $p^{\rm th}$ excited state is,

$$\langle\psi_{{\rm e},p}^\pm|\tilde{O}|\mathcal{C}_\alpha^\pm\rangle = \chi_{m,n}(t)e^{i\omega_{\rm r}(m-n)t}e^{-i(E_0-E_{{\rm e},p}^\pm)t}\langle\psi_{{\rm e},p}^\pm|a^{\dagger m}a^n|\mathcal{C}_\alpha^\pm\rangle$$
$$+ \chi_{m,n}(t)e^{i\omega_{\rm r}(n-m)t}e^{-i(E_0-E_{{\rm e},p}^\pm)t}\langle\psi_{{\rm e},p}^\pm|a^{\dagger n}a^m|\mathcal{C}_\alpha^\pm\rangle \tag{7}$$

In general, the matrix elements $\langle\psi_{{\rm e},p}^\pm|a^{\dagger m}a^n|\mathcal{C}_\alpha^\pm\rangle$ and $\langle\psi_{{\rm e},p}^\pm|a^{\dagger n}a^m|\mathcal{C}_\alpha^\pm\rangle$ can be non-zero because unlike $a$, the coherent states are not eigenstates of $a^\dagger$. Consequently, action of $a^{\dagger m}$ on $|\mathcal{C}_{\alpha e^{i\phi}}^\pm\rangle$ can cause direct transitions to excited states up to $p = m$ if the spectral density of the coupling is resonant with these transitions. To elaborate with the example, consider the oscillator ($\phi = 0$) driven in-phase by a classical coherent drive at frequency $\omega_1$ so that $O = \chi_0\cos(\omega_1 t)(e^{i\omega_{\rm r}t}a^\dagger + {\rm h.c})$. In the limit of large $\alpha$, $a^\dagger|\mathcal{C}_\alpha^\pm\rangle \sim \alpha|\mathcal{C}_\alpha^\mp\rangle + |\psi_{{\rm e},1}^\mp\rangle$ so that $\langle\psi_{{\rm e},1}^\mp|a^\dagger|\mathcal{C}_\alpha^\pm\rangle \sim 1$. In this case, a resonant excitation to $|\psi_{{\rm e},1}^\mp\rangle$ is possible if $\omega_1 = \omega_{\rm r} - (E_0 - E_{{\rm e},1}^\pm) \sim \omega_{\rm r} - 4K\alpha^2$, otherwise such transitions will not be resonant. Consequently, we see that a coherent drive at $\omega_1 = \omega_{\rm r}$ cannot cause resonant excitations out of the cat manifold and will only causes Rabi oscillations within the cat manifold. There will be some virtual excitations but these are negligible if $\chi_0 \ll |\omega_1 - \omega_{\rm r} + 4K\alpha^2| = 4K\alpha^2$. We can extend this analysis to show that a resonant beam-splitter coupling between Kerr-cat qubits, $H_{0,1} + H_{0,2} + \chi_0(a_1^\dagger a_2 + a_2^\dagger a_1)$, will lead to a $Z_1Z_2$ interaction between the qubits and can be used to realize a two-qubit $ZZ(\theta) = e^{i\theta Z_1Z_2/2}$ gate [11,14].

The analysis in this section can also be used to predict the structure of errors in the cat qubit. Suppose, the system operator that enters in the interaction Hamiltonian between the cat and bath is of the form $\chi_{m,n}a^{\dagger m}a^n + {\rm h.c.}$. Clearly, the primary effect of coupling with the environment will be $Z$ errors within the cat manifold and leakage out of the cat manifold since the $a^{\dagger m}$ term directly excites $|\mathcal{C}_{\alpha e^{i\phi}}^\pm\rangle$ up to the $m^{\rm th}$ excited manifold $|\psi_{{\rm e,m}}^\pm\rangle$. The probability of both of these errors increases polynomially with $\alpha$ where the polynomial depends on $m, n$. On the other hand, the coupling of the bath to the $X$

or $Y$ component of the cat qubit is exponentially suppressed [14]. The leakage can be autonomously corrected by dissipatively *cooling* the system to the stable cat states. This cooling can be achieved, for example, by two-photon dissipation since the cat states are the steady states of two-photon driven nonlinear oscillator in the presence of two-photon dissipation [14]. More importantly, as long as leakage remains confined to one of the quasi-degenerate eigenstates, its autonomous correction only leads to additional phase-flip errors. This is because in order to introduce a bit-flip type error, the degeneracy between the even- and odd-parity eigenstates comprising the Hilbert space must be lifted. However, the first $p$ excited states, with $p \leq \alpha^2/4$, are quasi-degenerate and consequently the bit-flips remain exponentially suppressed. Therefore, we find that the cat qubit has a biased noise channel where the bias increases exponentially with $\alpha$ [14]. The bias-noise property of the Kerr-cat qubit has been verified experimentally [15].

# 4 Bias-Preserving CX Gate

## 4.1 The X Gate

Now, recall from Eq. (3) that the orientation of the cat state in phase-space is defined by the phase $\phi$ of the two-photon drive. If this phase changes adiabatically from 0 to $\pi$, then the cat states $|\mathcal{C}_\alpha^\pm\rangle$ transform to $|\mathcal{C}_{-\alpha}^\pm\rangle = \pm|\mathcal{C}_\alpha^\pm\rangle$. Therefore, rotating the phase of the two-photon drive by $\pi$ is equivalent to a $X$ operation, because $X(x|\mathcal{C}_\alpha^+\rangle + y|\mathcal{C}_\alpha^-\rangle) = x|\mathcal{C}_\alpha^+\rangle - y|\mathcal{C}_\alpha^-\rangle$. Furthermore, from the discussion in the previous section, it follows that at time $t$ the predominant errors from the environment will be phase-flips or $Z_{\phi(t)}$-errors in the instantaneous basis of $H_0(\phi) = -Ka^{\dagger 2}a^2 + P(a^{\dagger 2}e^{2i\phi(t)} + a^2 e^{-2i\phi(t)})$, where $Z_{\phi(t)} = |\mathcal{C}_{\alpha e^{i\phi(t)}}^+\rangle\langle\mathcal{C}_{\alpha e^{i\phi(t)}}^-| + |\mathcal{C}_{\alpha e^{i\phi(t)}}^-\rangle\langle\mathcal{C}_{\alpha e^{i\phi(t)}}^+|$. In other words, the noise *rotates* along with the cat. Therefore, if a phase error occurs at time $t$, the state of the cat-qubit changes from $x|\mathcal{C}_{\alpha e^{i\phi(t)}}^+\rangle + y|\mathcal{C}_{\alpha e^{i\phi(t)}}^-\rangle$ to $x|\mathcal{C}_{\alpha e^{i\phi(t)}}^-\rangle + y|\mathcal{C}_{\alpha e^{i\phi(t)}}^+\rangle$. Finally, when $\phi(t) = \pi$, the state is

$$-x|\mathcal{C}_\alpha^-\rangle + y|\mathcal{C}_\alpha^+\rangle = XZ(x|\mathcal{C}_\alpha^+\rangle + y|\mathcal{C}_\alpha^-\rangle). \tag{8}$$

The above equation shows that a phase-flip error during the $X$ gate is equivalent to a phase-flip error before an ideal implementation of the gate.

## 4.2 The CX Gate

Based on the X gate, the idea of the CX gate is to rotate the target cat qubit in phase space conditioned on the state of the control cat qubit [14]. The desired Hamiltonian is,

$$\begin{aligned}
H_{\text{CX}} = & -K\left(a_{\text{c}}^{\dagger 2} - \alpha^2\right)\left(a_{\text{c}}^2 - \alpha^2\right) - \frac{\dot{\phi}(t)}{4\alpha}a_{\text{t}}^\dagger a_{\text{t}}(2\alpha - a_{\text{c}}^\dagger - a_{\text{c}}) \\
& -K\left[a_{\text{t}}^{\dagger 2} - \alpha^2 e^{-2i\phi(t)}\left(\frac{\alpha - a_{\text{c}}^\dagger}{2\alpha}\right) - \alpha^2\left(\frac{\alpha + a_{\text{c}}^\dagger}{2\alpha}\right)\right] \times \\
& \left[a_{\text{t}}^2 - \alpha^2 e^{2i\phi(t)}\left(\frac{\alpha - a_{\text{c}}}{2\alpha}\right) - \alpha^2\left(\frac{\alpha + a_{\text{c}}}{2\alpha}\right)\right].
\end{aligned} \tag{9}$$

Here, the subscripts c, t refer to the control and target cat respectively. The first term in the above expression is the Hamiltonian of the parametrically driven nonlinear oscillator stabilizing the control cat-qubit. To understand the other terms, recall from the previous section that $a_{\text{c}}^\dagger, a_{\text{c}} \sim \alpha Z_c$. Therefore, if the control qubit is in the state $|0\rangle$ ($\sim |\alpha\rangle$, for

large $\alpha$) then, to a good approximation, the above Hamiltonian is equivalent to

$$H_{\text{CX}}^{|0\rangle_c} \equiv - K \left( a_c^{\dagger 2} - \alpha^2 \right) \left( a_c^2 - \alpha^2 \right) - K \left( a_t^{\dagger 2} - \alpha^2 \right) \left( a_t^2 - \alpha^2 \right). \tag{10}$$

Consequently, when the control qubit is in the state $|0\rangle$ the state of the target oscillator remains unchanged. On the other hand, if the control qubit is in the state $|1\rangle$ ($\sim |-\alpha\rangle$, for large $\alpha$) then Eq. (9) is equivalent to

$$H_{\text{CX}}^{|1\rangle_c} \equiv - K \left( a_c^{\dagger 2} - \alpha^2 \right) \left( a_c^2 - \alpha^2 \right) - K \left( a_t^{\dagger 2} - \alpha^2 e^{-2i\phi(t)} \right) \left( a_t^2 - \alpha^2 e^{2i\phi(t)} \right) - \dot{\phi}(t) a_t^\dagger a_t. \tag{11}$$

From the second term of this expression we see that the cat states $|\mathcal{C}_{\alpha e^{i\phi(t)}}^{\pm}\rangle$ are the instantaneous eigenstates in the target oscillator. As a result, if the phase $\phi(t)$ changes adiabatically, respecting $\dot{\phi}(t) \ll |\Delta\omega_{\text{gap}}|$ then the orientation of the target cats follow $\phi(t)$ and $\alpha$ evolves in time to $\alpha e^{i\phi(t)}$. During this rotation in phase space the target cat also acquires a geometric phase $\Phi_g^{\pm}(t)$ proportional to the area under the phase space path, $e^{i\Phi_g^{\pm}(t)}|\mathcal{C}_{\alpha e^{i\phi(t)}}^{\pm}\rangle$ where $\Phi_g^{\pm}(t) = \phi(t)\alpha^2 r^{\mp 2}$. In the limit of large $\alpha$, the difference in the two geometric phases decreases exponentially in $\alpha^2$, $\Phi_g^- - \Phi_g^+ = 4\phi(t)\alpha^2 e^{-2\alpha^2}/(1 - e^{-4\alpha^2})$. Consequently, for large $\alpha$, the state $|1\rangle \otimes d_0|\mathcal{C}_\alpha^+\rangle + d_1|\mathcal{C}_\alpha^-\rangle$ evolves in time to $e^{i\Phi_g(t)}|1\rangle \otimes d_0|\mathcal{C}_{\alpha e^{i\phi(t)}}^+\rangle + d_1|\mathcal{C}_{\alpha e^{i\phi(t)}}^-\rangle$ where $\Phi_g(t) = \Phi_g^-(t) \sim \Phi_g^+(t)$. In other words, the geometric phase, effectively, is only an overall phase which results in an additional $Z_c(\Phi_g)$ rotation on the control qubit. This rotation can be accounted for in software or by an application of $Z_c(-\Phi_g)$ operation. Or it can be directly cancelled by the dynamic phase resulting from the third term in Eq. (11). The projection of this term in the cat basis is given by

$$P_{\mathcal{C},t}\dot{\phi}(t)a_t^\dagger a_t a_c P_{\mathcal{C},t} \sim \dot{\phi}(t)\alpha^2 \left[ r^2 |\mathcal{C}_{\alpha e^{i\phi(t)}}^+\rangle\langle\mathcal{C}_{\alpha e^{i\phi(t)}}^+| + r^{-2}|\mathcal{C}_{\alpha e^{i\phi(t)}}^-\rangle\langle\mathcal{C}_{\alpha e^{i\phi(t)}}^-| \right]. \tag{12}$$

As a result, we find that when the control cat is in state $|1\rangle$, an arbitrary state of the target qubit $d_0|\mathcal{C}_\alpha^+\rangle + d_1|\mathcal{C}_\alpha^-\rangle$ evolves in time to $d_0|\mathcal{C}_{\alpha e^{i\phi(t)}}^+\rangle + d_1|\mathcal{C}_{\alpha e^{i\phi(t)}}^-\rangle$. Consequently, the Hamiltonian in Eq. (9) leads to the evolution desired to implement the bias-preserving CX gate.

## 5    Summary and Conclusion

In this chapter we considered the outstanding challenge in improving quantum error correction when the noise is biased, namely realizing a bias-preserving CX gate. We studied the Kerr-cat qubit and its noise properties to show that it is a biased noise qubit. The dominant errors are phase-flips whereas, bit-flip errors are exponentially suppressed with the size of the cat. We saw how a bias-preserving CX gate can be realized with this qubit. With the availability of such a bias-preserving CX gate it becomes possible to exploit the structure of noise and gain practical advantage in quantum error correction [10].

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
