# Peer review of "Biased Noise Kerr-Cat Qubits for Fault-Tolerant Quantum Computing"

_SciPost Physics Lecture Notes_

## Round 1 · Referee Report · Jérémie Guillaud (Referee 1) · 2021-10-28

Strengths
1- Introduction of bias-preserving X and CX gates for the Kerr-cat qubit encoding 2- Clear exposure of the interest of Kerr-cat qubits for error correction
Report
This work introduces the "Kerr-cat qubit", a bosonic qubit engineered by applying a two-photon drive to a Kerr-nonlinear oscillator. This introduction focuses on the biased noise structure inherited from the specific encoding of the qubit into coherent states far apart in phase-space, and on the interest of this noise bias for quantum error correction. Recent theoretical studies have established the interest of using such biased-noise qubits for quantum error correction to reduce the overall overhead, by tailoring the error correction code to the specific noise structure of the underlying qubits.
Most importantly, this work introduces the working principle to realize some gates containing a Pauli X component (the X and CX gates), while preserving the structure of the noise. The ability to perform such gates in a bias-preserving manner is non-trivial and possible because of the bosonic nature of the Kerr-cat qubit. This is both original and of crucial importance to implement quantum error correction.
One minor point that could perhaps benefit from additional details is the analysis of the exponential suppression of bit-flips errors in section 4.2. It is argued that the leakage outside the code space induced by some error processes (typically those that contain an $\hat{a}^\dagger$ term) and its autonomous correction via some cooling process such as two-photon dissipation, cannot introduce non-exponentially suppressed bit-flip errors as long as only the first $p$ excited eigenstates of the Kerr-Hamiltonian are populated, with $p \leq \alpha^2/4$. This is because the energy degeneracy of the first $p$ pairs of excited eigenstates is exponentially suppressed with $\alpha^2$, such that the dephasing between the states of each pair (or equivalently, the resulting bit-flips on the codespace after the cooling process) is exponentially suppressed with $\alpha^2$.
From a practical point of view, this implies that, in order to remain in the regime where the bit-flips are exponentially suppressed in presence of error processes inducing leakage, the Kerr non-linearity cannot be made much larger than the rate of the cooling process, to ensure that only the first $p$ excited state are populated.
Given that the noise bias of the cat qubits, and the ability to tune this bias to very large values by increasing the mean number of photons, is a central assumption in many recent architectures proposed based on these qubits, I believe the paper would perhaps benefit from the addition of a discussion, or numerical evidence, that the typical Kerr-nonlinearity / two-photon dissipation rate achieved experimentally allow to access the regime of exponentially suppressed bit-flips.
Most importantly, this work introduces the working principle to realize some gates containing a Pauli X component (the X and CX gates), while preserving the structure of the noise. The ability to perform such gates in a bias-preserving manner is non-trivial and possible because of the bosonic nature of the Kerr-cat qubit. This is both original and of crucial importance to implement quantum error correction.
One minor point that could perhaps benefit from additional details is the analysis of the exponential suppression of bit-flips errors in section 4.2. It is argued that the leakage outside the code space induced by some error processes (typically those that contain an $\hat{a}^\dagger$ term) and its autonomous correction via some cooling process such as two-photon dissipation, cannot introduce non-exponentially suppressed bit-flip errors as long as only the first $p$ excited eigenstates of the Kerr-Hamiltonian are populated, with $p \leq \alpha^2/4$. This is because the energy degeneracy of the first $p$ pairs of excited eigenstates is exponentially suppressed with $\alpha^2$, such that the dephasing between the states of each pair (or equivalently, the resulting bit-flips on the codespace after the cooling process) is exponentially suppressed with $\alpha^2$.
From a practical point of view, this implies that, in order to remain in the regime where the bit-flips are exponentially suppressed in presence of error processes inducing leakage, the Kerr non-linearity cannot be made much larger than the rate of the cooling process, to ensure that only the first $p$ excited state are populated.
Given that the noise bias of the cat qubits, and the ability to tune this bias to very large values by increasing the mean number of photons, is a central assumption in many recent architectures proposed based on these qubits, I believe the paper would perhaps benefit from the addition of a discussion, or numerical evidence, that the typical Kerr-nonlinearity / two-photon dissipation rate achieved experimentally allow to access the regime of exponentially suppressed bit-flips.

---

## Editorial Decision

awaiting_resubmission